# Actin Cytoskeleton Dynamics and Type I IFN-Mediated Immune Response: A Dangerous Liaison in Cancer?

**DOI:** 10.3390/biology10090913

**Published:** 2021-09-14

**Authors:** Paola Trono, Annalisa Tocci, Martina Musella, Antonella Sistigu, Paola Nisticò

**Affiliations:** 1Institute of Biochemistry and Cell Biology, National Research Council, Via Ramarini 32, Monterotondo Scalo, 00015 Rome, Italy; paola.trono@cnr.it; 2Tumor Immunology and Immunotherapy Unit, IRCCS Regina Elena National Cancer Institute, Via Chianesi 53, 00144 Rome, Italy; annalisa.tocci@ifo.gov.it (A.T.); antonella.sistigu@gmail.com (A.S.); 3Dipartimento di Medicina e Chirurgia Traslazionale, Università Cattolica del Sacro Cuore, 00168 Rome, Italy; martinamusella90@gmail.com

**Keywords:** actin cytoskeleton, innate immune sensors, Type I IFN signaling, viral mimicry, therapy resistance

## Abstract

**Simple Summary:**

Actin cytoskeleton is a dynamic subcellular component critical for maintaining cell shape and for elaborating response to any stimulus converging on the cell. Cytoskeleton constantly interfaces with diverse cellular components and affects a wide range of processes important in homeostasis and disease. What has been clearly demonstrated to date is that pathogens modify and use host cytoskeleton to their advantage. What is now emerging is that in sterile conditions, when a chronic inflammation occurs as in cancer, the subversion of tissue homeostasis induces an alarm status which mimics infection. This activates cellular players similar to those that solve an infection, but their persistence may pave the way for tumor progression. Understanding molecular mechanisms engaged by cytoskeleton to induce this viral mimicry could improve our knowledge of processes governing tumor progression and resistance to therapy.

**Abstract:**

Chronic viral infection and cancer are closely inter-related and are both characterized by profound alteration of tissue homeostasis. The actin cytoskeleton dynamics highly participate in tissue homeostasis and act as a sensor leading to an immune-mediated anti-cancer and anti-viral response. Herein we highlight the crucial role of actin cytoskeleton dynamics in participating in a viral mimicry activation with profound effect in anti-tumor immune response. This still poorly explored field understands the cytoskeleton dynamics as a platform of complex signaling pathways which may regulate Type I IFN response in cancer. This emerging network needs to be elucidated to identify more effective anti-cancer strategies and to further advance the immuno-oncology field which has revolutionized the cancer treatment. For a progress to occur in this exciting arena we have to shed light on actin cytoskeleton related pathways and immune response. Herein we summarize the major findings, considering the double sword of the immune response and in particular the role of Type I IFN pathways in resistance to anti-cancer treatment.

## 1. Introduction

Tissue organization and homeostasis are based on a collection of circuits which include inflammatory signaling regulating tissue environment [1]. The unbalance of these connections may lead to the development of inflammatory diseases and immunosuppression as occurs in both chronic viral infections and cancer, with Type I Interferon (IFN-Is) as key drivers [2].

IFN-Is are produced by virally infected cells to induce a cell-intrinsic antiviral status in neighboring cells to resolve infection; however, prolonged IFN-I signaling leads to immune dysfunction, by inducing suppressive factors that hamper immunity to promote chronicity [2]. Similarly, in cancer, IFN-Is drive multiple mechanisms which promote inflammatory signals but may also initiate feedback suppression in both immune and cancer cells [2].

A crucial player involved in the maintenance of normal tissue architecture and function is the cell cytoskeleton: in polarized epithelia, cortical actin filaments underneath the plasma membrane form a complex network that preserves tight junction architecture and represents a physical barrier to the entry of pathogens, as demonstrated by host cytoskeleton disruption upon infection [3]. Viral infections are characterized by subversion of the actin cytoskeleton, sustained by viral proteins that either activate cell signaling pathways or directly recruit actin-remodeling proteins [4].

Similar to viral infection, cancer progression is governed by actin cytoskeleton modification of tumor cells, as clearly demonstrated during epithelial-mesenchymal transition (EMT) process. Epithelial cells undergo a cellular transformation and acquire motility and invasive ability, via invadopodia, lamellipodia, and filopodia, whose function strictly depends on the rearrangements of actin filaments [5]. A still growing body of evidence demonstrates that actin dynamics represent a platform for signaling pathways governing cancer progression [6], as evidenced during EMT. Similarly, new evidence is emerging showing that pathogens, to optimize infection, subvert the structure and function of the cytoskeleton which, after pathogen detection, activates the innate immune signaling pathways [4,7,8]. The actin cytoskeleton interacts with and regulates diverse innate immune proteins and, when microbial infection occurs, it provokes perturbations in cytoskeleton that are perceived as danger signals able to activate the innate immune response [9]. Cytoskeleton regulatory proteins identified as associated with immune sensors have also been reported as crucial in tumor progression and, by binding immune proteins, participate in the regulation of cell intrinsic innate immunity signaling and may suppress immune response during tumor progression.

Herein, we will firstly introduce mechanisms of IFN-I signaling in infection and cancer, then review evidence supporting the involvement of actin cytoskeleton proteins in the anti-viral Type I IFN signaling and the consequences occurring in cancer progression and resistance to therapy.

## 2. Mechanisms of IFN-I Signaling in Infection and Cancer

IFN-Is are the first line of defense against viral infection and central coordinators of inflammation within the tumor microenvironment (TME) [10]. They are a large family of small protein and glycoprotein cytokines, including diverse homologous IFN-α subtypes (11 in mice and 13 in humans), a single IFN-β protein, and the less investigated IFN-ε, IFN-τ, IFN-κ, IFN-ω, IFN-δ, and IFN-ζ, all of which bind to the same heterodimeric IFN-α/β receptor (IFNAR) composed of the IFNAR1 and IFNAR2 chains [11]. IFN-Is are encoded by intronless genes which are clustered together on chromosome 4 in mice and on chromosome 9 in humans [12]. IFN-Is are secreted by all nucleated cells upon engagement of diverse families of heterologous receptors, so called pattern recognition receptors (PRRs), by pathogen-associated molecular patterns (PAMPs), during infection [13], and damage-associated molecular patterns (DAMPs), in cancer (Figure 1) [14,15,16]. PRRs are surface, cytosolic, and endosolic innate immune receptors [17] which include (i) transmembrane Toll-like receptors (TLRs), (ii) endosolic TLRs, (iii) cytosolic RIG-I-like receptor (RLRs), (iv) NOD-like receptors (NLRs), and (*v*) DNA sensors [17]. IFN-I induction and regulation have recently been extensively reviewed in [11] and are not the focus of this review.

TLRs sense microbial lipoproteins (TLR1, TLR2, and TLR6), lipopolysaccharide (TLR4), flagellin (TLR5), and microbial and self-damaged nucleic acids as double-stranded- and single-stranded-RNAs (TLR3, TLR7, and TLR8), and DNA (TLR9) [18]. RLRs include retinoic acid-inducible gene (RIG)-I, melanoma differentiation-associated protein (MDA)5, and laboratory of genetics and physiology (LGP)2, all sensing pathogenic and damaged RNAs [19,20]. NLRs encompass more than 20 cytoplasmic PRRs known to detect components of the bacterial cell wall such as peptidoglycan and to form a specialized multiprotein complex, the inflammasome [21,22]. DNA sensors include TLR9, the DNA-dependent activator of IRFs (DAI), the RNA polymerase-III (POLR3), the cyclic guanosine monophosphate–adenosine monophosphate (cGAMP) synthase, the PYHIN proteins, namely the absent in melanoma (AIM)2, and IFN-γ-inducible (IFI)16, and diverse DDX and DHX helicases [23,24,25,26,27,28,29]. Interestingly, a crosstalk between RNA and DNA sensing pathways occurs and can potentiate efficient antiviral responses [30].

Through different adaptor molecules (e.g., TIR-domain containing adaptor inducing IFN (TRIF), MyD88 adapter-like (Mal), mitochondrial anti-viral signaling protein (MAVS) and stimulator of IFN genes (STING), among the others), these engaged PRRs activate the IkB kinase ε (IKKε)/TANK-binding kinase 1 (TBK1) complex, and thus IFN-regulatory factors (IRF)3, IRF7, and nuclear factor kappa-light-chain-enhancer of activated B cells (NF-kB), which results in two subsequent waves of production of IFN-β and IFN-α subtypes, respectively (Figure 1) [31].

Once released, IFN-Is act in an autocrine, paracrine, and systemic manner, by binding IFNAR and thus inducing the activation of the Janus kinase (JAK)—signal transducer and activator of transcription (STAT) pathway. The signal transduction downstream of the canonical JAK/STAT activation triggers the transcription of hundreds of different IFN-stimulated genes (ISGs), which are responsible for the various and pleiotropic range of cellular responses to viruses and virus-unrelated malignancies (Figure 1) [11]. The spectrum and kinetic of ISG mainly rely on which IFN-I subtype initiates the signaling cascade [32]. Indeed, each IFN-I subtype induces a unique set of ISGs [33]. As a result, different IFN-I subtypes induce tissue- and context-dependent cellular responses during infection, cancer, and inflammation [33]. For a comprehensive view of ISGs refer to http://www.interferome.org (accessed on 30 July 2021), http://www.innatedb.com (accessed on 30 July 2021), and http:/www.niaid.nih.gov (accessed on 30 July 2021).

Multiple cell types within the tumor microenvironment (TME), including cancer, immune, and stromal cells, produce and respond to IFN-Is [11,34]; however, most studies describe ISGs within the tumor bulk not distinguishing between expression among the cell populations. Despite their source, IFN-Is and their related signatures, were shown to exert both tumor promoting and host protective roles, by directly acting on cancer cells and indirectly affecting stromal and immune cell status and function [35]. This knowledge has far-reaching implications in tumor immunology and clinical oncology, as strategies targeting IFN-I signatures could be paramount to develop more effective anticancer (immune)therapies.

Similar to what occurs after infection by a virus, aberrant nucleic acids activate DNA and/or RNA PRRs and this activation results in production of IFNs and the induction of interferon-stimulated genes (ISGs). This viral mimicry has been reported to occur after genotoxic therapies [14,36] and accumulating evidence have sparked interest towards this process in tumor immunity.

As extensively reported, different pathogens induce host actin cytoskeleton modifications, which act as innate immune sensors. A greater understanding of the mechanisms contributing to the regulation of IFN-I signaling will provide clinically relevant diagnostic and therapeutic tools.

Below a general introduction to actin cytoskeleton is outlined, followed by details of interactions between cytoskeleton dynamics and innate immunity components.

## 3. Actin Cytoskeleton: Components, Dynamics, and Emerging Role in Innate Immunity

Actin, along with microtubules, intermediate filaments, and septins, represents major cytoskeletal components of vertebrate cells, constantly undergoing a dynamic reorganization. Actin exists in the cell as monomeric globular actin (G-actin) and polymeric filamentous actin (F-actin). An initiation complex stabilizes actin monomers and initiates polymerization, thus filament formation occurs through the action of proteins such as profilin and cortactin, while depolymerization is achieved by proteins such as cofilin or gelsolin [37]. Actin filaments can bundle to form larger structures, interacting with other actin binding proteins, such as fascins, or can be crosslinked by branching, by actin nucleating proteins [37]. Under the control of regulatory proteins affecting all aspects of actin filament dynamics, actin filaments can be arranged to form a wide range of structures. Stress fibers are large networks of actin filaments that can span nearly the entire length of the cell and their association with myosin enables contractility. Cortical actin is a network of actin filaments that resides underneath the plasma membrane. Depending on organization, actin filaments may form cellular extensions, such as lamellipodia, filopodia, microvilli, and large membrane ruffles [38].

Actin binding proteins act as messengers by regulating fundamental signaling pathways and by communicating to the genome in healthy and diseases including cancer. Notably, actin dynamics have also been linked to gene transcription, and different mechanisms that communicate the cytoplasmic actin status to the nuclear genome have been described [6].

The presence of actin in the nucleus has been clearly shown, although its functional role is still debated [39]. Recently actin has been linked to gene expression regulation and enhances Polymerase II clustering after serum stimulation or IFNγ treatment [40]. Furthermore it has been reported that chromatin regulation is essential in interferon and innate anti-viral gene expression [41], suggesting that nuclear actin may have a role in interferon-stimulated gene regulation.

The development of actin modulating drugs is a field of interest in cancer and it is becoming evident that actin-binding proteins (ABPs) may represent a target to regulate actin polymerization and depolymerization, as reported for the small-molecule inhibitors of Ena/VASP EVH1 interaction, able to impair invasion and extravasation of breast cancer cells [42]. Notably, ROCK1/ROCK2 inhibitor Y-276432, which prevents MDA-MB-231 breast cancer cell metastasis, and MRCK inhibitor BDP5290 which impairs human squamous cell carcinoma invasion, have been investigated [43,44]. However, actin cytoskeletal dynamics as cancer therapeutic targets are not the focus of this review and are excellently reviewed elsewhere [5].

A growing body of evidence has revealed the role of the cytoskeleton as a structural determinant of cell-autonomous host defense, by detecting bacterial pathogens and favoring the antibacterial responses [8]. Bacterial pathogens manipulate host cytoskeletal proteins to promote their intracellular replication and survival [7] and similarly viruses subvert host actin cytoskeleton to optimize viral replication and virion production [4].

Thus, it is not surprising that a number of associations between cytoskeletal related proteins and immune sensors of pathogens have been described (Figure 2). The recruitment of immune molecules to cytoskeletal structures can be viewed as a strategy to prevent immune response in non-infected cells, by restraining immune activation during resting conditions. Vice versa, cytoskeleton plasticity allows a fast release of immune molecules in specific contexts.

The notion that cytoskeleton and its regulators play key roles in immune defense is well-recognized in plants, where, according to the “guard hypothesis”, the cytoskeleton acts as a guarded structure, whose integrity is strictly monitored and any perturbation by pathogens triggers a defense response [45]. In plant, cytoskeleton acts as a scaffold for many plasma membrane-associated proteins, interacts with numerous membrane receptors associated with immune signaling and plant defense signaling, and participates in the re-localization of immune effectors to the sites of infection by regulating intracellular trafficking. Plant cytoskeleton is therefore one of the major regulator of innate immunity signaling, representing the stage from which immune-associated processes are mobilized and oriented, controlling the movement of organelles, proteins, and biochemical signals involved in the defense signaling [46].

Innate immune sensors and effectors have historically been thought to function only in the context of pathogen infection, however they have been reported to control host homeostatic activities in absence of pathogen threat.

Interestingly, a recent work investigated the regulation of immune genes in the three major types of structural cells: epithelium, endothelium and fibroblasts, and identified an epigenetically encoded immune potential in structural cells under tissue homeostasis, which was triggered in response to systemic viral infection [47]. Thus, in structural cells there is a widespread expression of immune regulators and cytokine signaling molecules, which activate immune gene activation for a rapid immunological response, as occurs in a viral infection and in the context of the TME [47].

In cancer cells during tumor progression, actin cytoskeleton rearrangements constantly occur and could be perceived as an alteration of homeostasis, which resembles the “alarm status” induced by pathogens. Afterward, cancer cells engage immune molecules associated to cytoskeleton to trigger a viral mimicry response which may provide a survival advantage to tumor progression.

## 4. Cytoskeleton Interactions with Components of Innate Immunity

Cell resistance to pathogens is achieved by both basal and inducible immune defenses. Mechanical features of the cell provide structural defense which represents the major component of the basal resistance, with the actin cytoskeleton as central player of this process [48].

Cellular actin cytoskeleton status and altered mechanical features of the cell participate to innate immunity activation with effects in the susceptibility to the infection, as emerges from the study of Irving et al., showing that Protein Kinase R (PKR) regulates actin dynamics via gelsolin (GSN), and enforces basal innate immune defense [48]. The antiviral PKR is an effector molecule of the innate immune system that responds to environmental stresses to regulate protein synthesis [49]. Critical for all antiviral and antiproliferative activities of PKR is its ability to phosphorylate EIF2α (Eukaryotic translation initiation factor 2 subunit 1) and to modulate several signaling pathways [50]. Its involvement in the regulation of a great number of cellular processes also accounts for both tumor-suppressive and tumor-stimulatory functions attributed to PKR [51]. PKR, not previously linked to control of the cell cytoskeleton, has been recently shown to bind GSN under homeostatic conditions and in turn sequesters it in an inactive conformation, reducing its association with actin (Figure 2f) [48]. GSN is a constitutively expressed protein that severs actin filaments and caps their barbed ends, affecting cell morphology, migration, invasion, and movement of the cell membrane [52]. The inhibition of GSN by PKR limits pathogen entry into cells, indicating a role for GSN in the susceptibility of cells to infection [48]. Indeed PKR, by inhibiting GSN-mediated severing of actin filaments blocks potential actin-dependent routes of viral entry [48]. Following infection with viruses, PKR is activated by double stranded viral RNA and GSN is released. Thus, active PKR can takes on its “classical” role in viral defense, inhibiting protein translation and enhancing PRR signaling [53].

The actin binding protein Filamin A, able to cross-link actin filaments, acts as a scaffold connecting the actin cytoskeleton with membrane receptors and signaling molecules [54]. Similarly to GSN, Filamin A has been reported to prevent viral entry by virtue of its actin-binding ability [55] and participates in the establishment and replication of diverse viruses by binding viral receptors. Malathi et al. have additionally demonstrated that Filamin A interacts with a component of host innate immune response, the endoribonulease RNase L (Figure 2e) [55]. IFN treatment, similarly to viral infection, activates RNase L, which cleaves single-stranded viral and host RNAs, leading to the activation of the RIG-I-like helicases RIG-I and MDA5 to amplify the production of IFN-β. In uninfected cells, Filamin A, by virtue of its actin-binding ability, may sequester a pool of RNase L in the cytoskeleton preventing viral entry by maintaining a barrier. Viral infections alter actin dynamics, and favor the release of RNase L, which can fuel antiviral signaling [55].

Different evidence demonstrates that components of the actin cytoskeleton regulate immune signaling by affecting the redistribution of innate immune-associated components necessary to initiate signaling and acting as scaffolding molecules to bridge components of the innate immune system [9,56].

NOD2, one of the major members of the NLRC subfamily of NLRs detects muramyl dipeptide (MDP) present in bacterial peptidoglycan and initiates NF-κB-dependent and mitogen-activated protein kinase (MAPK)-dependent gene transcription [57]. Different evidence supports a role for NOD2 in cancer where it has been reported as a critical component of gastric epithelial barrier homeostasis [57] and a number of NOD agonists have demonstrated immunostimulatory and anticancer activity [58]. The actin cytoskeleton modulates NOD2-dependent NF-κB signaling, as identified by Legrand-Poels et al., in intestinal epithelial cancer cells and in normal fibroblasts with ectopic expression of NOD2 [59]. NOD2 is sequestered in actin cytoskeleton structures such as lamellipodia and membrane ruffles in association with Ras-related C3 botulinum toxin substrate 1 (Rac1), a GTP-binding Rho family member, that when activated drives cell protrusion by inducing local actin rearrangements [60]. Actin disruption releases NOD2 from cytoskeleton structures and increases NF-κB activation and IL-8 secretion (Figure 2g), indicating that the recruitment of NOD2 in Rac1-induced dynamic cytoskeletal structures may represent a strategy to prevent the NF-κB signaling in uninfected cells and to rapidly activate NOD2 during bacterial infection [59].

Other evidence on the sensing of Rho GTPase activation state by NOD2 is excellently reviewed in [61] and collectively suggests that Rho GTPase activity links actin cytoskeletal remodeling to innate immune responses mediated by NOD1 and NOD2.

Multiple studies have established a crosstalk between signaling via the innate immune receptors and the mitochondria, placing mitochondria as a crucial platform for innate immunity [62,63,64]. Interactions of mitochondria with the cytoskeleton are crucial for normal mitochondrial function and for their localization within cells [65].

The involvement of actin-related proteins in the distribution and localization of immune sensors at mitochondria has been reported [56,66].

The focal adhesion kinase FAK is a protein tyrosine kinase located primarily at focal adhesions (FAs), where a direct physical link between intracellular actin cytoskeleton and the surrounding ECM occurs. FAK participates along with integrins to the transmission of signals between the ECM and the cytoplasm [67]. In an outstanding study from Bozym et al. the response to virus infection involves FAK, which functionally links actin cytoskeletal perturbations to IFN-Is [56]. Starting with the observation that FAK deficiency enhances RNA virus replication and impairs NF-κB and IFN-β signaling, Bozym et al. demonstrated that, in response to virus infection, FAK is recruited to the mitochondrial membrane, where interacts with MAVS to enhance NF-kB- and IFN-β-mediated antiviral signaling (Figure 2a) [56]. The authors propose that FAK acts as a scaffold to recruit the signaling and cytoskeletal components to the mitochondrial membrane required for MAVS re-localization, and this results in the enhancement of RLR signaling. Although the mechanism is still not completely elucidated, FAK integrates actin cytoskeletal changes to the alterations in mitochondrial morphology, that accompany MAVS-mediated innate immune signaling. It is noteworthy that coxsackievirus B (CVB) is able to cleave FAK, generating two distinct fragments, with one of them acting as a dominant-negative inhibitor of FAK-mediated antiviral signaling indicating that the virus targets FAK as a mechanism to evade host immunity [56]. Similarly, in cancer, FAK inhibition hampers the composition an immunosuppressive TME in pancreatic cancer and sensitizes tumors to immunotherapy [68].

Fascin, the main actin-bundling protein in filopodia and invadopodia, promotes cell motility, invasion, and adhesion through its canonical actin bundling function [69]. Notably Fascin interacts with the Linker of Nucleoskeleton and Cytoskeleton (LINC) complex, and localizes at the nucleus to regulate nuclear actin [69]. High levels of Fascin have been found in many types of metastatic tumors and correlate with clinically aggressive phenotypes, poor prognosis, and shorter survival [70]. Interestingly, Matsumura et al. have identified Fascin1 as a suppressor of the RIG-I signaling pathway in colon cancer cells, where Fascin1 impairs IFN-β production by interfering with the function of IKK (Figure 2d) [71].

A bridge between components of the innate immune system and actin binding proteins has been proposed by Kouwaki et al. for zyxin which acts as a scaffold between RLRs and MAVS [66]. Zyxin is a LIM-domain protein that regulates assembly of the actin cytoskeleton, first identified as a protein localized at adhesion plaques and at the ends of actin filaments [72], and now recognized as a crucial element of the mechano-transducing system at FAs where facilitates force-induced actin polymerization [73]. Zyxin also shuttles into the nucleus, with an important role in the regulation of cell differentiation, proliferation, adhesion and migration [74]. To the best of our knowledge, the study of Kouwaki et al. is the first to report a function of zyxin in antiviral innate immune response. The authors show that zyxin and MAVS partially co-localize at mitochondria, and physical interactions between RLRs and MAVS are abrogated by *ZYX* knockdown, thereby reducing MAVS-mediated signaling (Figure 2c) [66]. Vice versa, the ectopic expression of zyxin augments MAVS-mediated IFN-β promoter activation, suggesting that zyxin stabilizes physical interactions between RLRs and MAVS, finally promoting IFN-I response.

The ability of immune molecules to sense the modulation of the actin cytoskeleton is crucial for RIG-I/MAVS signaling activation, as evidenced by Ohman et al. [75]. They showed that actin and RIG-I/MAVS signaling components translocate from the cytoplasm to mitochondria upon Influenza A virus infection of human primary macrophages. Notably, cytochalasin D, the inhibitor of actin polymerization that disrupts actin microfilaments, clearly inhibits influenza A virus-induced expression of IFNs and TNF-α [75]. Thus, actin filaments have been proposed to serve as tracks for RIG-I to mitochondria, suggesting that intact actin cytoskeleton structure is crucial for antiviral response [75]. A direct association between actin and RIG-I clearly emerges from the study of Mukherjee et al. [76], which demonstrated that RIG-I is concentrated at sites of actin-rich membrane ruffles in epithelial cells (Figure 2b). Notably, RIG-I associates with the apical tight junction (TJ) complex where it strongly co-localizes with ZO-1 in both cultured intestinal epithelial cells (IEC) and human colon biopsies. Of note, cytochalasin D-mediated depolymerization of the actin cytoskeleton in polarized IEC provokes the rapid relocalization of RIG-I from the junctional complex to actin-enriched clusters, leading to the induction of IFN-I signaling [76]. Thus, RIG-I not only acts as an important sensor for the detection of viral infection but also plays a role in actin cytoskeletal regulation and is involved in a mechanism of innate immune recognition, particularly in the polarized IECs of the gastrointestinal tract [76].

Among the proteins regulating the assembly and geometry of actin networks, Ena/VASP proteins localize to actin-rich structures, including the leading edge of lamellipodia and the tips of filopodia, stress fibers, and focal adhesions [77,78].

Ena/VASP proteins share a modular domain organization, where distinct regions harbor different binding domains that enable Ena/VASP proteins to interact with a number of proteins, including zyxin and profilin [77]. MENA (mammalian ENA homolog, ENAH), along with VASP (vasodilator-stimulated phosphoprotein), and EVL (Ena-VASP-like), belong to the family [77]. Multiple splice variants of ENAH gene have been identified and are involved in several mechanisms critical for cancer progression [79,80,81,82,83]. ENAH contains a small coding exon 11a that is included only in epithelial cells by Epithelial Splicing Regulatory Proteins 1 and 2 (ESRP1 and ESRP2) [84,85]. The exon 11a localizes between the F-actin binding and tetramerization regions in EVH2 and harbors two putative phosphorylation sites closed to actin binding domains, suggesting that hMENA^11a^-specific phosphorylation might alter its effects on actin polymerization [84]. In invasive tumor cells, the hMENA^11a^ down-regulation occurs, and the human MENA (hMENA) splice variant lacking the internal exon 6 (hMENAΔv6) is up-regulated [79,80,82]. Our group has demonstrated that, while hMENAΔv6 sustains β1 integrin expression and activation, overexpression of hMENA^11a^ inhibits β1 integrin and FAK activation [80]. Low hMENA^11a^ expression, along with high overall hMENA, is a pre-requisite of EMT, perturbs cell-cell junction integrity, and identifies PDAC and early NSCLC patients and with poor prognosis [80,86].

The analysis of the transcriptomic profile of hMENA^11a^ depleted (si-11a) NSCLC cells revealed that hMENA^11a^ loss leads to the enrichment of several transcripts related to IFN-I signaling, including IFNB1. Notably, si-11a cells produce IFN-β and are characterized by the activation of JAK/STAT1 pathway. However, both IFN-β production and STAT1 activation are impaired by the depletion of RIG-I, leading to the hypothesis that the depletion of hMENA^11a^, that induces a deep rearrangement of the actin cytoskeleton, might activate the actin associated viral sensor RIG-I, mimicking a viral infection (manuscript in preparation), again supporting the link between actin cytoskeleton, RIG-I, and IFN-I signaling.

## 5. Opposing Effects of IFN-Is in Disease Progression and Resistance to Therapy

As above delineated, actin cytoskeleton may act as a platform of cell signaling able to regulate type I immunity. In this section we report how IFN-Is may have, when chronically secreted, a contribution in cancer progression and resistance to therapy.

IFN-Is are essential drivers of antitumor immunity, able to stimulate an efficient anti-tumor immune-mediated response, as extensively demonstrated when IFN signaling primes T cell response [15,87]. Qualitative and quantitative differences have to be considered to define IFN signaling properties and the timing and duration of the signaling are critical for innate immune control of virus replication, as reported for simian immunodeficiency virus (SIV). Indeed, repeated IFNα 2a administration decreased anti-viral gene expression enabling increased SIV reservoir size [88]. During persistent viral infections, an elevated IFN signature correlates with disease progression, indicating that IFN-I signaling pathway may represent a therapeutic target as shown for lymphocytic choriomeningitis virus (LCMV). The authors demonstrated that that IFN-I signaling contributes to tissue disorganization, as shown by IFN-I blockade which preserved the splenic architecture [89].

As occurs in chronic viral infection, certain types of tumors become resistant to IFN-I anti-proliferative and apoptotic effects, due to mutational events in IFN signaling, which involve not only modification in IFNAR expression, but also genetic alteration in JAK-STAT pathway, thus reducing the efficacy of IFN-I -based immunotherapy [90,91,92]. The crucial role of IFN signaling dysregulation in therapy resistance mechanisms has been defined in preclinical and clinical cancer models, as here discussed. Dunn and colleagues showed that LNCaP human prostate adenocarcinoma cell line is unresponsive to IFN-I and -II due to the lack of expression of JAK1 [91]. In human leukemic H123 Jurkat cell line the loss of STAT2 has been associated with resistance to IFN-α induced apoptosis [92]. A functional antagonism between STAT1 and STAT5 has been identified in melanoma cells with STAT1 able to inhibit, and STAT5 able to promote tumor growth. Furthermore, STAT5 overexpression in the nucleus not only contributes to progression of nevi lesions to primary and metastatic melanoma, but also to IFN-α resistance [93].

Among the negative regulators of IFN signaling pathway, the suppressor of cytokine signaling (SOCS) proteins have been demonstrated, when overexpressed, as able to protect neuroendocrine tumor cells from IFN-I-mediated apoptosis. Conversely SOCS1 silencing resulted in increased and prolonged STAT1 phosphorylation levels, leading to apoptotic death of cancer cells [94].

Anti-tumor immunity requires organized and spatially defined interaction among TME cell components, with the IFN-I as a major orchestrator of immune environment in cancer. In both human and mouse models of colorectal cancers (CRC), Katlinski and colleagues demonstrated that IFNAR1 loss plays a key role in the establishment of an intra- tumoral immune privileged niche in the CRC TME and may be predictive of CRC patient outcomes [95]. Notably, the IFNAR1 down regulation on cytotoxic T lymphocytes (CTL) hampers their survival within the TME and limits the efficacy of CAR-based therapy in solid tumors [95].

In the new scenario of anticancer immune therapies, including adaptive transfer of tumor infiltrating lymphocytes (TILs) and immune check point blockade (ICB), mechanisms of resistance have been linked to genetic alteration in IFNAR signaling and antigen processing and presenting machinery genes [96,97]. The overexpression of the transcription factor FOXA1, which directly binds the STAT2 DNA-binding domain (DBD), impedes the STAT2 related expression of IFN-signaling and antigen presentation machinery (APM) genes. The authors suggest that FOXA1 overexpression could be a predictor of therapy resistance to immune- and chemotherapy in prostate, breast, and bladder cancer patients [98].

Searching for the effect of anti PD-1 blockade immunotherapy on cancer genomic evolution, Zaretsky et al. found that melanoma patients treated with anti-PD1 acquired mutations in the genes related to IFN and antigen presentation pathways [97]. Indeed they found by whole genome sequencing, that two of four melanoma metastatic patients analyzed acquired loss of function mutation in the genes coding for JAK1/2, and the third patient presented a truncated mutation in the gene coding for beta-2-microglobulin (B2M) [97]. In agreement JAK1/2 mutations contributes to an immunoediting process that presumably occurs in patients that will be not responsive to PD-1 blockade therapy [99].

As in anti-PD-1 immunotherapy, resistance to anti-CTLA4 treatment in melanoma is driven by homozigous deletion (HD) of IFN-I and defensin activated genes which are related to oncogenic and cell-cycle pathways and are able to inhibit immunoresponsive pathways [100].

With the advance and success of cancer immunotherapy we are gathering an increasing number of data that may contribute to understanding the IFN-I role in the different tumor contexts. Data are emerging on IFN-induced negative regulatory pathways as barriers to the effectiveness of ICB therapy, as recently highlighted by Benci and colleagues. The authors unveiled novel mechanisms of therapy resistance, influenced by the nature and persistence of IFN signaling in the TME. Prolonged IFN signaling enhanced STAT1-associated open chromatin, and leads to the expression of interferon-stimulated genes in tumor cells and induction of multiple T cell inhibitory receptor ligands. Moreover, the authors suggest that IFN sustains adaptive resistance by involving multiple pathways either PD-L1 dependent or PD-L1-independent [101]. In agreement with the role of sustained IFN-I signaling as a mechanism of resistance to PD-1 blockade in melanoma patients, the L. Zitvogel lab demonstrated that the activation of IFNAR1 signaling increases the expression of Nitric oxide synthase 2 (NOS2) on tumor cells and leucocytes cell fractions leading to intratumor accumulation of regulatory T cells and myeloid cells [16]. Chronic tumor intrinsic IFN production renders the cancer cells more prone to the aberrant accumulation of double stranded RNA (dsRNA) mediated by increased levels of the MDA5, RIG-I, and PKR sensors. This ISG signature in tumors render cancer cells susceptible to the loss of the ds-RNA enzyme ADAR (adenosine deaminase acting on RNA), which has been proposed a therapeutic target in patients possessing a high ISG signature [102].

The activation of IFNAR1/STAT1 signaling in head and neck squamous cell carcinoma (HNSCC) by IFN-α is associated with the establishment of an immunosuppressive microenvironment favoring the expression of PD-L1 in HNSCC cells, PD-1 in immune cells and impairs the lithic activity of NK cells [103].

Among the effective combined therapies with ICB, radiation therapy (RT) may contribute to an efficient antitumor immune response, through induction of immune cell death and IFN secretion [104,105]. However, it has been reported that activation of IFN-I signaling in cancer cells may also protect irradiated tumors from CD8^+^ T cell–mediated cytotoxicity by regulating *Serpinb9* [106].

Furthermore, IFN-mediated mechanisms of resistance have been also reported for standard of care treatment in breast cancer, such as aromatase inhibitors (AIs), that have been shown able to induce the overexpression of ISGs in cancer cells [107]. A recent work identified an association with high IFN signaling and reduced sensitivity to CDK4/6 inhibitors which is associated with an immunosuppressive index and poor outcome in patients, suggesting that IFN signature could be relevant to stratify patients to be treated with endocrine therapy alone or combined with cdk4/6 blockade [108].

Finally, the activation of IFN signaling pathway contributes also to tumor resistance to oncolytic virotherapy as indicated by the beneficial effects of combining virotherapy with IFN inhibitors. Dold and colleagues demonstrated that IFN signaling is up-regulated in ovarian cancer cell lines, making them resistant to oncolytic virus infection and destruction. Importantly, the authors show that the co-administration of the replication-competent vesicular stomatitis virus pseudotyped with the glycoprotein of the lymphocytic choriomeningitis virus (VSV-GP) and ruxolitinib, a JAK1/2-ihnibitor, enhances the oncolytic activity of VSV-GP in vitro and prevents tumor recurrence in intraperitoneal/orthotopic ovarian cancer xenograft mice model [109].

Similar to what was observed in ovarian cancer, malignant peripheral nerve sheath tumor (MPNST) cells are resistant to the infection of oncolytic herpes simplex virus (oHSV) through the activation of the JAK/STAT1 pathway which leads to the up regulation of ISGs. Importantly the administration of ruxolitinib reduces ISGs expression in MPNST cells rendering them more sensitive to viral replication and spread [110].

Collectively these data highlight the opposing roles of IFNs and reveal that the timing and duration of exposure to IFNs may exert an activation or suppression of the immune response, strongly impacting the clinical outcome (Figure 3, Table 1). With the advance in the immunotherapy field it is likely that more suppressive mechanisms will be elucidated, along with the role of actin cytoskeleton in the complex IFN-I networks, to promote new effective therapies in cancer.

## 6. Conclusions

Herein, we highlighted how actin cytoskeleton regulatory proteins interface with critical players of IFN-I signaling during homeostasis and how the deregulation of actin cytoskeleton, occurring in tumor progression as well as in chronic infections, might affect these interactions, paving the way for tumor growth. The comprehension of the molecular mechanisms underlying these processes will contribute to further understanding the complexity of IFN-I signaling in the tumor context.

## Figures and Tables

**Figure 1 biology-10-00913-f001:**
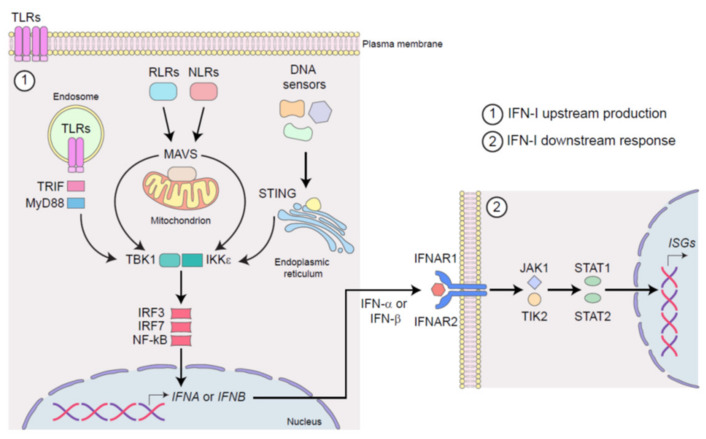
Schematic representation of IFN-I signaling in infection and cancer. (**1**) The stimulation of surface, cytosolic, and endosomal pattern recognition receptors (PRRs), by pathogen-associated molecular patterns (PAMPs) during infection or by damage-associated molecular patterns (DAMPs) in cancer, induces IFN-I production. Upon activation, Toll-like receptors (TLRs) signals through TRIF and MyD88, RIG-I-like receptor (RLRs) and NOD-like receptors (NLRs) through MAVS, whereas DNA sensors engage STING. All of these adaptor molecules activate IKKε/TBK1 kinase complex, which in turn triggers the activation of transcription factors IRF3/7 and NF-κB, leading to the production of IFN-α and IFN-β. (**2**) Once released from infected or cancerous cells, IFN-α and IFN-β act in an autocrine, paracrine, and systemic manner by binding the heterodimeric receptor IFNAR1-2, which in turn activates JAK/STAT pathway leading to the transcription of multiple IFN-stimulated genes (ISGs).

**Figure 2 biology-10-00913-f002:**
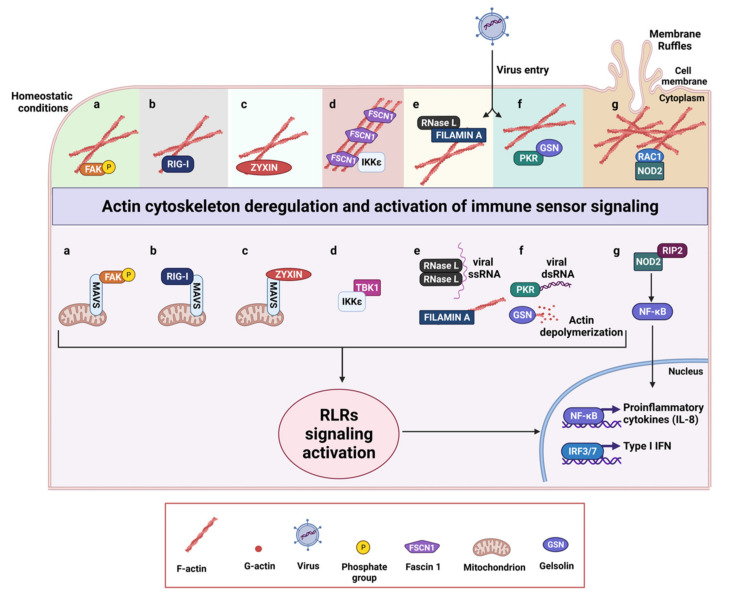
Schematic representation of actin regulatory proteins and their interactions with components of innate immunity. Under homeostatic conditions multiple innate immune sensors are directly or indirectly linked to the actin cytoskeleton. Deregulation of actin cytoskeleton occurring during tumor progression or following viral infections leads to the activation of RLR signaling activation. (**a**) Focal adhesions kinase (FAK) associates with the actin cytoskeleton in resting cells, but, upon actin reorganization due to virus entry, FAK relocalizes to the mitochondria where it interacts with MAVS, acting as a scaffold between immune signaling and actin cytoskeleton at the mitochondrial membrane. (**b**) The antiviral sensor RIG-I directly binds F-actin by the N-terminal caspase activation and recruitment domains (CARDs) in resting cells and rearrangements of the actin cytoskeleton lead to its rapid relocalization to the mitochondria, where it interacts with MAVS. (**c**) Zyxin partially co-localizes with MAVS at mitochondria, where it sustains and stabilizes physical interactions between RLRs and MAVS by its N-terminal domain. (**d**) Fascin 1 (FSCN1) constitutively binds IKKε, restraining RIG-I signaling pathway. Upon actin reorganization, IKKε is released and associates with TBK1 which in turn phosphorylates IRF3. (**e**) In uninfected cells, Filamin A sequesters a pool of inactive RNase L in the cytoskeleton preventing viral entry by maintaining a barrier. Viral infection alters actin cytoskeleton and activates RNase L, which cleaves single-stranded viral and host RNAs, leading to the activation of the antiviral response by RIG-I and MDA5. (**f**) PKR binds gelsolin (GSN) under homeostatic conditions and in turn sequesters it in an inactive conformation, reducing its association with actin. Upon virus entry, PKR is activated by double stranded viral RNA, releases GSN, which can promote actin depolymerization, and activates RLRs signaling. (**g**) NOD2 binds RAC-1 and co-localizes with F-actin in membrane ruffles. Upon actin cytoskeleton rearrangements such as membrane ruffles disruption, NOD2 is released and can interact with RIP2 leading to NF-kB activation and IL-8 production. The activation of RLR signaling culminates with the production of type I IFNs and inflammatory cytokines.

**Figure 3 biology-10-00913-f003:**
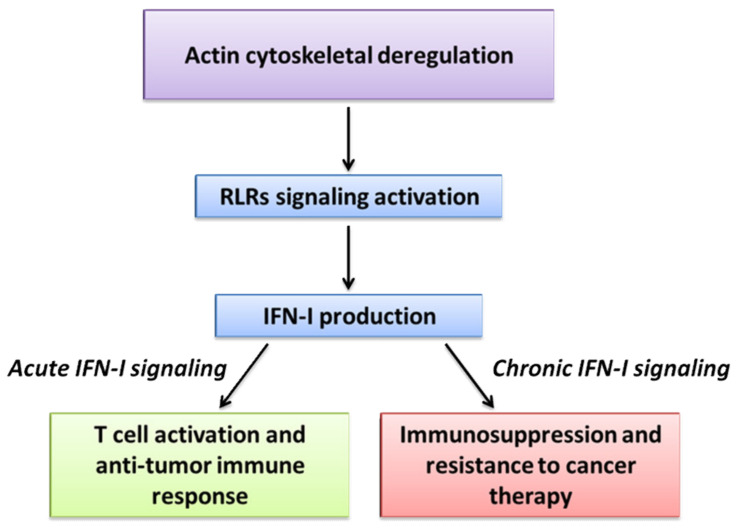
Actin cytoskeletal deregulation and viral mimicry in cancer. In cancer cells actin cytoskeleton rearrangements constantly occur during tumor progression and could be perceived as an alteration of homeostasis that resembles a viral infection. This viral mimicry induces RLRs signaling activation, culminating with IFN-I production. Depending on the context, acute or chronic IFN-I signaling may have opposite effects in cancer.

**Table 1 biology-10-00913-t001:** Effects of acute and chronic IFN signaling in cancer.

Acute IFN-I Signaling *	Chronic IFN-I Signaling
Apoptotic and antiproliferative effect [111]	Resistance to immune-checkpoint blockade therapy [16,103]
Induction of cell differentiation [112,113]	Resistance to radiation therapy [106]
Induction of senescence [113]	Resistance to oncolytic viral therapy [109,110]
Up regulation of antigen presenting and processing machinery genes [114]	Resistance to aromatase inhibitors [107]
	Resistance to CDK4/6 inhibitors [108]

* Some references recapitulating the different mechanisms involved in anti-tumor immune response mediated by acute IFN signaling.

## Data Availability

Not applicable.

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
