# Peer review of "Actin Cytoskeleton Dynamics and Type I IFN-Mediated Immune Response: A Dangerous Liaison in Cancer?"

_biology, 2021, doi:10.3390/biology10090913_

Round 1

Reviewer 1 Report

This is an interesting review describing the underlying mechanisms of actin cytoskeleton during tumor progression. However, I have a few comments:

1. The organization of the entire review does not flow smoothly. Each section needs to be reorganized so that every paragraphs in each section is connected.

2. Figure 1: Please provide more information in the figure legend for readers convenience without having to refer to the main text.

3. Lines 283-285: "Multiple studies... [54]" Please include more than one reference. Also, lines 260-263: ""different evidences..." please provide more than one evidence.

4. Please ensure references are cited in main text. For instance, Line 333: Please provide the reference, no reference was provided in Line 209 that states "A recent work".

5. All figures: please make sure they are properly referred to in the main text. For instance, Figure 1 has been mentioned as FIgure 1(a), FIgure 1(b).... However, there is only FIgure 1 but no Figure 1a,1b,.... 

6. It is not easy to follow or understand throughout the manuscript. Intensive editing of English language and style is highly recommended.

Author Response

  1. The organization of the entire review does not flow smoothly. Each section needs to be reorganized so that every paragraphs in each section is connected.

Response 1: We thank the Reviewer for his/her comments. To render the review smoother, we have modified the text by adding a few sentences of connection in each section.

  1. Figure 1: Please provide more information in the figure legend for readers convenience without having to refer to the main text.

Response 2: According to Reviewer’s suggestion we have added more information in the figure legend.

  1. Lines 283-285: "Multiple studies... [54]" Please include more than one reference. Also, lines 260-263: ""different evidences..." please provide more than one evidence.

Response 3:  Thanks for suggesting to add references that accordingly we have included.

  1. Please ensure references are cited in main text. For instance, Line 333: Please provide the reference, no reference was provided in Line 209 that states "A recent work".

Response 4:  We have provided the missed reference.

  1. All figures: please make sure they are properly referred to in the main text. For instance, Figure 1 has been mentioned as FIgure 1(a), FIgure 1(b).... However, there is only FIgure 1 but no Figure 1a,1b,.... 

Response 5: We feel that the format of Figure 1 legend has led to a misunderstanding. The figure includes panel from a to g that are extensively described in the Figure legends. Please note that in the revised version, the previous Figure 1 has been now Figure 2, since  Reviewer 2 suggested to add another Figure.  

  1. It is not easy to follow or understand throughout the manuscript. Intensive editing of English language and style is highly recommended.

Response 6: In agreement with the Reviewer’s suggestion, the manuscript has been carefully revised by a native English speaker.

Reviewer 2 Report

Trono et al. provide an overview of actin cytoskeleton modification strategies utilized by viruses that are mimicked by cancer, leading to type I IFN response and its importance in anti-cancer treatment. The review is well written and organized, however, Figure 1 has still the potential for visual improvement. The molecular mechanism of actin cytoskeleton damage may open potential drug candidates to treat cancer. There are few concerns to improve the review-

  1. Visual presentation always attracts readers- For subtitle: mechanism of IFN-I signaling in infection and cancer, please summarize in a figure what is written.
  2. Figure 1 font size is small, use appropriate font size with 300 dpi image.
  3. Actin cytoskeleton stabilizing proteins can be an alternative way to treat cancers? What is common with viral infections? and how it differs in IFN response?
  4. A brief description between cytoplasmic and nuclear actin dynamics differences and similarities that affect viral and cancer diseases? Do they have roles in IFN-I response modulation? It will increase the impact of the review, however, the authors cover ref. 59, 60, and 61.
  5. Actin cytoskeleton also acts as a barrier to viral infection, how it is different from cancer progression? add a paragraph
  6. Is there any treatment strategy based on modulation of actin cytoskeleton dynamics at the molecular level. It will be interesting to know if any potential drug targeting cytoskeleton regulatory proteins are known and summarize in this review.

Author Response

  1. Visual presentation always attracts readers- For subtitle: mechanisms of IFN-I signaling in infection and cancer, please summarize in a figure what is written.

Firstly we thank the Reviewer to consider our review as important in the field  and for his/her constructive criticisms that have improved the paper.

Response 1: In this revised version, we have included a new Figure, Figure 1, which illustrates mechanisms of IFN-I signaling in infection and cancer.

  1. Figure 1 font size is small, use appropriate font size with 300 dpi image.

Response 2: We thank the Reviewer for this suggestion and we have accordingly modified the figure.

  1. Actin cytoskeleton stabilizing proteins can be an alternative way to treat cancers? What is common with viral infections? and how it differs in IFN response?

Response 3: We have discussed the option to treat cancer by stabilizing actin cytoskeleton proteins by adding sentences and relative references (from line 207 to 222; references 42, 43, 44) in the paragraph: “Actin cytoskeleton: components, dynamics and emerging role in innate immunity”

  1. A brief description between cytoplasmic and nuclear actin dynamics differences and similarities that affect viral and cancer diseases? Do they have roles in IFN-I response modulation? It will increase the impact of the review, however, the authors cover ref. 59, 60, and 61.

Response 4: In agreement with the Reviewer’s suggestions, we have added new sentences relative to nuclear actin dynamics in the paragraph: “Actin cytoskeleton: components, dynamics and emerging role in innate immunity” (from line 201 to 206; references 39, 40,41).

  1. Actin cytoskeleton also acts as a barrier to viral infection, how it is different from cancer progression? add a paragraph

Response 5: Thanks for raising this concern. We have added in the Introduction section (from line 52 to 62; references 3,5) a paragraph on the role of actin cytoskeleton as a barrier in viral infection and how the epithelial barrier may be deregulated in cancer, pointing on Epithelial-Mesenchymal transition.

  1. Is there any treatment strategy based on modulation of actin cytoskeleton dynamics at the molecular level. It will be interesting to know if any potential drug targeting cytoskeleton regulatory proteins are known and summarize in this review.

Response 6:  Although the actin cytoskeleton regulatory proteins as druggable target is not the focus of this review we have appreciated Reviewer’s suggestion and added sentences relative to drug targeting cytoskeleton regulatory proteins in the paragraph: “Actin cytoskeleton: components, dynamics and emerging role in innate immunity” (from line 207 to 222; references 42, 43, 44). 

Round 2

Reviewer 1 Report

The organization of the manuscript looks much better in the second revision. However, there are still some minor comments that should be address:

  1. Figure 2 and 3: What do you mean by Figure 21 and Figure 32?
  2. Also, there are blue and red highlighted words. What are the difference?